# OLIGOTREND, a global database of multi-decadal chlorophyll-a and water quality time series for rivers, lakes and estuaries

Camille Minaudo[1,2], Andras Abonyi[3,4,5], Carles Alcaraz[6], Jacob Diamond[7], Nicholas J.K. Howden[8,9], Michael Rode[10,11], Estela Romero[12,1], Vincent Thieu[13], Fred Worrall[14], Qian Zhang[15], Xavier Benito[6,1]

[1]Departament de Biologia Evolutiva, Ecologia i Ciències Ambientals, Universitat de Barcelona (UB), Diagonal 643, 08028 Barcelona, Spain
[2]Institut de Recerca de la Biodiversitat (IRBio), Universitat de Barcelona (UB), Diagonal 643, 08028 Barcelona, Spain
[3]MTA-ÖK Lendület Fluvial Ecology Research Group, Karolina Street 29, H-1113 Budapest, Hungary
[4]HUN-REN Centre for Ecological Research, Institute of Aquatic Ecology, Karolina Street 29, H-1113 Budapest, Hungary
[5]WasserCluster Lunz – Biologische Station GmbH, Dr. Carl Kupelwieser Promenade 5, A-3293 Lunz am See, Austria
[6]Marine and Continental Waters Program, Institute of Agrifood Research and Technology (IRTA), 43540 La Ràpita, Catalonia, Spain
[7]Intergovernmental Hydrological Programme, UNESCO, Paris, France, 7 Place de Fontentoy 75015
[8]School of Civil, Aerospace and Design Engineering, University of Bristol, Bristol, BS8 1TR, UK
[9]Cabot Institute, University of Bristol, Bristol, BS5 9LT, UK
[10]Department of Aquatic Ecosystem Analysis and Management, Helmholtz Centre for Environmental Research - UFZ, Magdeburg 39104, Germany
[11]Institute of Environmental Science and Geography, University of Potsdam, Potsdam 14476, Germany
[12]Global Ecology Unit, Centre for Ecological Research and Forestry Applications (CREAF), Campus UAB, Bellaterra, Spain
[13]Sorbonne Université, CNRS, EPHE, UMR 7619 METIS, 4 place Jussieu, Box 105, 75005, Paris, France
[14]Department of Earth Sciences, University of Durham, Durham, UK
[15]University of Maryland Center for Environmental Science. U.S. Environmental Protection Agency Chesapeake Bay Program, 1750 Forest Drive, Suite 130, Annapolis, MD 21401, United States

*Correspondence to*: Camille Minaudo (camille.minaudo@ub.edu)

**Abstract.** Reversed eutrophication, called oligotrophication, has widely been documented globally over the last 30 years in rivers, lakes, and estuaries. However, the absence of a comprehensive and harmonized dataset has hindered a deeper understanding of its ecological consequences. To address this data gap, we developed the OLIGOTREND database, which
contains multi-decadal time series of chlorophyll-a, nutrients (nitrogen and phosphorus), and related physicochemical parameters, totalling 4.3 million observations. These data originate from 1,894 unique monitoring locations across estuaries (n = 238), lakes (687), and rivers (969). Most time series covered the period 1986–2022 and comprised at least 15 years of chlorophyll-a observations. Each location is associated to catchment and hydroclimatic attributes. Trend and breakpoint analyses were applied to all time series. Chlorophyll-a showed temporally variable and ecosystem-specific responses to
nutrient declines with an overall declining trend for 18% of the time series, contrasting greatly with a majority of declining trends for nutrient concentrations. We harmonized the database to ensure reproducibility, ease of access, and support future updates and contributions. Available at https://doi.org/10.6073/pasta/a7ad060a4dbc4e7dfcb763a794506524 (Minaudo and Benito, 2024) the OLIGOTREND database supports collaborative efforts aimed at further advancing our understanding of

biogeochemical and biological mechanisms underlining oligotrophication, and ecological impacts of global long-term
environmental change.

**Short summary.** Many waterbodies undergo nutrient decline globally, called oligotrophication, but a comprehensive dataset to understand ecosystem responses is lacking. The OLIGOTREND database comprises multi-decadal chlorophyll-*a* and nutrient time series from rivers, lakes, and estuaries with 4.3 million observations from 1,894 unique measurement locations.
The database provides empirical evidence for oligotrophication responses with a spatial and temporal coverage exceeding previous efforts.

**Introduction**

Decades of freshwater and estuarine eutrophication in the 20th century spurred coordinated national efforts to reduce aquatic nutrient loads and subsequent algal blooms (Pinay et al., 2017). The most effective actions have included improved wastewater
collection and treatment, better coordinated watershed management, and the regulation of phosphorus in detergents (Conley et al., 2009; Némery and Garnier, 2016). Evidence from rivers, lakes, and estuaries already suggests that such efforts can indeed reverse eutrophication at time scales ranging from months to years and decades, in a process termed oligotrophication or re-oligotrophication. However, our understanding of oligotrophication is still incomplete (Anneville et al., 2019; Hoyer et al., 2002; Ibáñez and Peñuelas, 2019), and the magnitude, direction, and timing of ecological responses to water quality
improvements remain to be better detected and quantified.  Declines in nutrients often coincide with a transition in primary producers in terms of quantity and community composition. The most reported change in inland and estuarine ecosystems is the systematic replacement of phytoplankton by submerged macrophytes (Ibáñez and Peñuelas, 2019). However, these shifts can follow nonlinear trajectories, typically explained by the occurrence of alternative stable states in lakes (Scheffer and Carpenter, 2003), rivers (Verdonschot et al., 2013), and estuaries (Duarte et al., 2009; Elliott and Quintino, 2007). Additional
complexities in predicting primary producer shifts arise due to nutrient legacies in the landscape that can create lags in ecosystem response (Van Meter et al., 2021; Stackpoole et al., 2019), and the presence of dams and weirs that alter the spatiotemporal variability of nutrient mobilization and transport (Zeng et al., 2023). Indeed, a wide range of contrasting trends in nutrients and primary production (as indicated by chlorophyll-a [*chla*]) are possible (Greening and Janicki, 2006; Kronvang et al., 2005; Murphy et al., 2022), including natural causes such as forest growth (Nilsson et al., 2024). Due to the complexity
of ecosystem responses to watershed nutrient reduction, a common predictive framework remains elusive, highlighting the need for cross-ecosystem analysis of oligotrophication trends.
Available water quality datasets, while plentiful, remain heterogeneous and often irregularly collected and reported, hindering their use in across-system studies. Moreover, oligotrophication has been primarily focused on local and regional-scale studies (e.g. Abonyi et al., 2018; Greening et al., 2014; Minaudo et al., 2021; Sabel et al., 2020) and isolated aquatic ecosystems. Thus,
the spatial extent of oligotrophication trends remain poorly constrained, and we lack an understanding of the connectivity of

oligotrophication responses across the watershed to estuary continuum. Even the best available harmonized, large-scale water quality databases commonly exclude *chla* (e.g., GRQA, Virro et al., 2021), limiting their utility to evaluate oligotrophication. Likewise, some databases may cover large numbers of observations, but exclude parallel measurements of *chla* and nutrients, mainly phosphorus (Nilsson et al., 2024; Spaulding et al., 2024) or are temporally limited relative to oligotrophication
timescales (Brehob et al., 2024). Therefore, there is a clear need for a centralized database of paired nutrient and primary producer observations at oligotrophication-relevant timescales across different ecosystems.

Here we present OLIGOTREND (Minaudo and Benito, 2024), a database of 4.3 million quality assessed public and open access observations of water quality variables and *chla* from rivers, lakes and reservoirs, estuaries and coastal bays, enabling the joint assessment of multi-decadal oligotrophication trends across spatial scales. We collected and harmonized multi-
decadal time series to facilitate its structure and reuse. The database also covers geospatial data, including catchment and waterbody attributes, climate variables, and a robust trend analysis of all water quality time series. Here we highlight some of the main findings from our first analyses of the database and describe possible research directions that OLIGOTREND holds the potential to answer.

## 2. Data and Methods

We followed a transparent and reproducible approach to produce the OLIGOTREND database, in line with best practices for Open Science in Ecology (Powers and Hampton, 2019). In particular, the entire data processing pipeline (Figure 1) was developed collaboratively in a version control GitLab repository (https://gitlab.com/OLIGOTREND/wp1-unify). Data are referenced according to their level ("L") in the processing pipeline. Time series extracted from various sources were defined as "L0a", preserving the original data structure and formatting. Time series were then harmonized ("L0b"), and a selection of
variables of interest (see Section 2.1) at sampling sites with at least 15 years of *chla* data qualified for the data quality assessment and check (QA/QC, see Section 2.2) and to be matched with geospatial data (see Section 2.3). Harmonized and curated time series together with catchment and waterbody attributes constitute "L1" data, i.e., analysis- and sharing-ready data. Any additional processing of L1 data, e.g. trend analyses, was considered as "L2" (see Section 2.4).

### 2.1. Data collection

In-situ *chla* concentrations and physicochemical parameters were extracted from open-source international, national, and regional water quality databases (Table 1). We first obtained data from queries to the Earth System Science Data portal (https://www.earth-system-science-data.net/), the Environmental Data Initiative repository (https://edirepository.org/), and the Scientific Data portal (https://www-nature-com.sire.ub.edu/sdata/). We then conducted a literature search on Web of Science (https://www.webofscience.com/wos/) and Scopus (https://www.scopus.com/) for further existing long-term *chla* and nutrient
time series. To do so, we used the following search terms: "TITLE or ABSTRACT (oligotrophication, reoligotrophication, chlorophyll, timeseries); and in TITLE or ABSTRACT (lake, river, estuary, coastal, estuarine); and in EVERYTHING (trend,

long term, multi-decadal)". When public and accessible, we directly extracted the datasets and proceeded with data harmonization. The database architecture (Figure 1) allows researchers to easily complement it with additional time series in the future. New additions to the database will be eased by a set of scripts available in a dedicated version control GitLab repository (https://gitlab.com/OLIGOTREND/wp1-unify) allowing to reproduce, update or add more timeseries from level L0a to higher data levels and products.

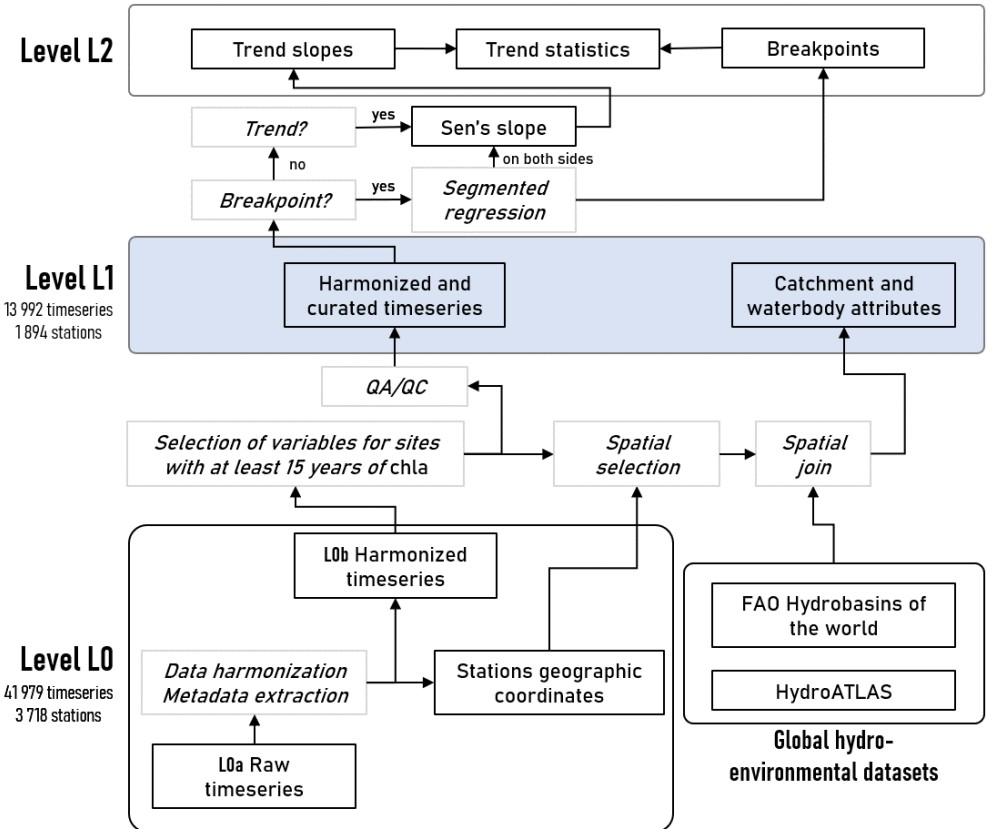

**Figure 1.** Data levels and procedure followed to produce the OLIGOTREND database, an ensemble of harmonized and curated time series of *chla* and water quality paired with catchment and waterbody attributes. QA/QC stands for quality assessment and quality check.

We gathered data as raw measurements, i.e. unprocessed or non-aggregated time series, and defined herein this data as level L0a. Extracted variables included chlorophyll-a (*chla*), water temperature (*wtemp*), conductivity (*cond*), pH, dissolved oxygen as concentration (*o2*) and percentage of saturation (*o2sat*), dissolved inorganic nitrogen (*din*), nitrate (*no3*), nitrate + nitrite (*no23*), ammonium nitrogen (*nh4*), Kjeldahl nitrogen (*nkjel*), total nitrogen (*tn*), orthophosphate or soluble reactive phosphorus (*po4*), total phosphorus (*tp*), dissolved organic carbon (*doc*), and total suspended solids (*tss*). The ecosystem types covered in this database included lakes and reservoirs, rivers, estuaries and coastal bays.

We primarily targeted databases identified with long periods of records without any filter on geographic location (Table 1). We discarded *chla* datasets obtained with remote sensing techniques, to ensure a strict comparability among observations. For stratifying deep lakes, we extracted values either for the euphotic layer, or from the upper 10 m if euphotic depth was unavailable, to avoid using data from light-limited conditions.

**Table 1. Data sources of the OLIGOTREND database.**

| Source | Link to data (and date of extraction when appropriate) | Spatial coverage |
|---|---|---|
| naiades French water quality portal | https://naiades.eaufrance.fr/ (last accessed 07/05/2024) | French national territory |
| Naderian et al., 2024 | https://doi.org/10.1016/j.resconrec.2023.107401 | Global |
| Chesapeake Bay Program | https://www.chesapeakebay.net/what/downloads/cbp-water-quality-database-1984-present (last accessed 30/01/2024) | Chesapeake Bay and watershed |
| LAGOS-NE | https://doi.org/10.1093/gigascience/gix101 | North-East USA |
| UK Harmonized Monitoring Dataset | https://datamap.gov.wales/documents/2633 (last accessed 17/06/2024) | England and Wales |
| Lake PCI | https://doi.org/10.20383/102.0488 | Temperate and cold northern lakes |
| Danish monitoring program | https://odaforalle.au.dk/login.aspx (last accessed 14/06/2024) | Denmark |
| Sacramento Bay Interagency monitoring | https://doi.org/10.6073/pasta/f58f8217c18f469e7fd565997a47813c | Sacramento-San Joaquin Delta (USA) |
| Elbe monitoring program | https://www.fgg-elbe.de/fachinformationssystem.html (last accessed 12/12/2023) | Elbe River watershed and estuary (Germany) |
| Filazzola et al., 2020 | https://doi.org/10.1038/s41597-020-00648-2 | Global |
| USGS-NWIS Data Retrieval | https://doi.org/10.5066/P9X4L3GE (last accessed 19/12/2023) | USA |
| GEMStat | https://gemstat.org/ (last accessed 11/06/2024) | Global |
| LTER Florida Everglades | https://doi.org/10.6073/pasta/f45fbf88dcf1f78f0d74c1dbdaaa8c7d | Florida Everglades (USA) |
| Danube River public program (HUN-REN CER, IAE) | https://doi.org/10.1111/fwb.13084 | Middle section of the Danube River (N-Budapest, Hungary) |
| Victoria State Government | http://www.data.water.vic.gov.au/ (last accessed 17/05/2024) | Victoria State (Australia) |
| Commission pour la Protection des Eaux du Léman (CIPEL) | https://www.cipel.org/en/ (last accessed 03/02/2023) | Lake Geneva, France-Switzerland |
| Ebro River monitoring program | https://doi.org/10.1016/j.scitotenv.2011.11.059 | Ebro River at Tortosa (Spain) |
| Romero et al., 2013 | https://doi.org/10.1007/s10533-012-9778-0 | Southwestern Europe |

## 2.2. Data harmonization and quality control

First, L0a time series were individually reformatted into standard units and data matrix headers, forming an ensemble of time series defined here as level L0b. Nutrient concentrations were expressed as mg $L^{-1}$ except *chla*, which remained in µg $L^{-1}$. Time series were named with a unique identifier (*uniquID*) per site corresponding to the concatenation of the following data separated by underscores: "ecosystem type", "basin", "station ID", e.g., "river_loire_04000100". Basin names were derived

from site geographic coordinates and the corresponding watershed according to the FAO dataset (Major hydrological basins of the world, 2025). Ecosystem type was either "estuary", "lake" or "river", corresponding to estuary or coastal bay, lake or reservoir, and river, respectively. The "station ID" was the one provided by the original data source. For each sampling site, the geographic coordinates found in the original metadata were used to create a point shapefile labelled with the station unique identifier (*uniquID*) as explained above. Stations with no geographic coordinates were discarded from the database.

Data quality was assessed and checked for all L0b time series from sampling stations presenting at least 15 years of *chla* data. The resulting dataset comprises the OLIGOTREND L1 data level (Figure 1). We did not remove any data in response to data curation (QA/QC) to allow users to design their own quality check procedure. Instead, we flagged potentially anomalous or suspicious observations. Valid observations were indicated with flag = 0. Quality control identified missing values (flag = 1), possible outliers (flag = 2), and abnormally repetitive values (flag = 3). Observations were considered as outliers when the corresponding values exceeded 3 times the interquartile range defined by site. Observations were considered abnormally repetitive when, at a given site and for a given variable, the corresponding value appeared more than 5 % of the time in the time series, not necessarily consecutively. Obvious mistakes in the units found in the original datasets at level L0b were identified and corrected by plotting the density of distribution of observed concentrations and scatter plots by pairs of variables (e.g., *chla* vs *tp*, *tp* vs *po4*, ...etc.) throughout the database.

### 2.3. Link with watershed and ecosystem properties

We linked inland sampling stations with the global HydroATLAS database (Lehner et al., 2022; Linke et al., 2019). The HydroATLAS has three distinct datasets: BasinATLAS, RiverATLAS, and LakeATLAS which represent sub-basin delineations (polygons), the river network (lines), and lake shorelines (polygons), respectively. Although we proceeded with the spatial join between HydroATLAS and OLIGOTREND stations, we acknowledge there may be a potential temporal mismatch between HydroATLAS properties and OLIGOTREND temporal coverage. Yet, we considered this spatial join would succeed at demonstrating the great variability of watershed and ecosystem properties encountered in the OLIGOTREND database.

First, we linked all OLIGOTREND sampling stations to the BasinATLAS by spatial selection of polygons of sub-basins (Pfafstetter level 12, i.e., the highest hierarchical sub-basin level in the BasinATLAS), overlapping with the point shapefile of L1 OLIGOTREND stations. A selection of watershed properties related to their physiography, climate, land cover, hydrology and anthropogenic pressures were extracted and linked to each station present in the database at the L1 level and intersecting with one of the BasinATLAS sub-basins. Similarly, the intersection of LakeATLAS lake polygons with L1 stations provided an ensemble of lake characteristics for 61% of the lake stations (418 out of 687). Finally, OLIGOTREND L1 river stations were linked to the RiverATLAS database by identifying the three nearest river segments using the function *joinbynearest()* in QGIS 3.26.2. For each possible station-segment match, the distance between the station and each segment was calculated, and the quality of the spatial join was assessed using a flagging system: if the distance to the nearest segment exceeded 500 m, a flag (flag = 1) was raised, indicating that the distance might be too large for the join to be considered valid. If the distance to

the second or third nearest segment was less than 10% greater than the distance to the nearest segment, a flag (flag = 2) was raised indicating that several river segments could potentially be selected. In that case, if these segments were associated with multiple sub-basins (HYBAS_L12 in HydroATLAS documentation), a flag value of 2.1 was set. If these segments were linked to multiple drainage basins (MAIN_RIV in HydroRIVERS), a flag value of 2.2 was set. All other associations identified during the spatial join were considered as valid, and flag value was set to flag = 0. Only stations with flag = 0 were considered reliable. Overall, out of 924 river stations, 90% were considered as valid. We found that 6.1% of stations were more than 500 m away from the closest HydroRIVERS segment, and 3.9% shown possible multiple associations (flag ≥ 2), sometimes with different sub-basins (1.3%, flag = 2.1) or drainage basins (0.3%, flag = 2.2). We acknowledge that there is some uncertainty in the spatial join between OLIGOTREND river stations and HydroRIVERS given the spatial resolution of the HydroSHEDS (15 arc-second). This uncertainty could be reduced by using a river network derived from a higher-resolution Digital Elevation Model. Stations with unmatched basin, lake or river segment from the HydroATLAS database were not removed from the OLIGOTREND database, but we did not account for them in the statistics and description of watershed attributes.

## 2.4. Time series metrics and trend analysis

We described the OLIGOTREND time series based on multiple metrics. These included the number of observations by each variable, the extent of the period of record, as well as the median, average and standard deviation of all valid values over the entire time series.

As a first step into the trend analysis, we quantified the proportion of time series showing lower annual averages in the second half of the time series compared to the first one. We chose annual averages over growing season averages to increase robustness in the metric because sampling frequency was sometimes unequally distributed seasonally. This further simplified the question of how to identify the growing season among sites across latitudes. We considered that a lower average value in the 2nd half of the time series indicated decline, regardless of the level of trend-complexity found in the time series.

A breakpoint and segmented regression analysis was performed using the R package *segmented* (Fasola et al., 2018). Whenever the Davies test (Davies, 1987) did not identify any non-constant linear regressions in time series, we conducted a Mann-Kendall trend analysis on annual averages with the R package *trend* (Pohlert, 2023). When the Mann-Kendall test detected a monotonic trend ($p < 0.01$), we calculated a Sen's slope over the complete dataset. Whenever the Davies test identified non-constant linear regressions, we fitted a segmented regression to the data with two joined segments, and the position of the temporal breakpoint and the corresponding interval estimation were identified. The Sen's slope was then quantified for both sides of the given breakpoint. For each segment, there were three possible trend types: declining, no trend, rising, noted as "-", "0" and "+", respectively. The combination of two joined segments or a single segment only when no breakpoint was detected provided a total of 12 possible trend types: "-","--", "+-", "0-", "-0", "0","00", "+0", "-+","0+","+", "++". We acknowledge a segmented regression with one breakpoint unlikely captures all the variety in trend patterns, but it may provide a comprehensive first assessment for non-linear and non-monotonic temporal patterns, robust enough to provide a first

overview on multi-decadal temporal trajectories. Outputs from the trend analysis and above-described statistical descriptors constitute level L2 data.

## 3. Database characteristics

### 3.1. Time series characteristics

We collected L0 data from 3,718 sampling stations, producing a total of 41,979 time series. Among these, 1,894 stations had at least *chla* for over 15 years and were selected for quality check and harmonization at level L1 (Figure 1). Following quality check, the OLIGOTREND database includes 4.3 million observations. Across all variables and time series, 83,807 observations (1.7 % of total observations) were flagged as outliers, and 691,000 (13.7 % of total observations) as repetitive observations. The highest proportion of abnormally repetitive observations were found for *nh4* and *tp* (34 % and 21 % of the observations, respectively, Table 2), likely related to detection and/or quantification limits above the actual concentrations. For *chla*, 13 % of the observations were flagged as repetitive (9.9%) or extreme outliers (3.4%). We only included the valid data points for all subsequent analysis and time series descriptions. Most L1 time series were multi-decadal with a median time series length of 33 years (Table 2).

The majority of *chla* time series included 5 observations per year (Table 2); only 16 % of time series were based on monthly sampling. We counted that 95% of *chla* time series exceeded 15 years, and 75%, 43% and 11% covered 20, 30 and 40 years, respectively. The longest *chla* time series covering more than 45 years originated from the LakePCI dataset (10 lake *chla* time series located in Sweden), the UK Harmonized Monitoring Program (42 rivers in England and Wales), and the Sacramento Bay Interagency monitoring (13 stations in estuarine area).

**Table 2. Overview of L1 data and percentage of data points flagged as invalid for each of the main variables. Ranges are presented as "median (10th percentile– 90th percentile)". The percentage of flagged observations (last column) correspond to possible outliers and abnormally repetitive values.**

| Variable | Number of time series | Time series length [yr] | Number of individual years covered | Number of observations | Frequency [observations/yr-1] | % of flagged observations |
|---|---|---|---|---|---|---|
| *chla* | 1885 | 29 (16-41) | 22 (15-36) | 158 (58-463) | 5 (3-14) | 13.3 |
| *cond* | 783 | 36 (20-43) | 31 (18-42) | 270 (168-527) | 8 (5-13) | 1.1 |
| *din* | 207 | 34 (15-35) | 35 (16-36) | 429 (176-588) | 12 (11-17) | 1.7 |
| *doc* | 157 | 23 (14-35) | 22 (15-35) | 267 (147-550) | 11 (7-21) | 2.8 |
| *nh4* | 916 | 33 (16-43) | 26 (15-42) | 139 (54-344) | 4 (2-10) | 38.1 |
| *nkjel* | 654 | 30 (15-43) | 23 (12-35) | 104 (31-221) | 3 (1-6) | 57.8 |
| *no23* | 176 | 22 (16-43) | 20 (11-34) | 188 (36-480) | 7 (2-14) | 18.9 |
| *no3* | 1008 | 34 (19-43) | 30 (17-42) | 245 (138-453) | 8 (4-12) | 4.8 |
| *o2* | 1005 | 35 (21-42) | 33 (18-42) | 302 (179-567) | 10 (5-15) | 0.8 |
| *o2sat* | 997 | 35 (21-42) | 33 (18-42) | 299 (182-557) | 10 (5-15) | 1.5 |

| | | | | | |
|---|---|---|---|---|---|
| ph | 1028 | 34 (17-42) | 28 (16-38) | 130 (64-377) | 4 (2-11) | 45.5 |
| po4 | 1014 | 34 (19-43) | 29 (17-42) | 218 (87-422) | 7 (3-11) | 20.5 |
| tn | 434 | 32 (17-37) | 24 (16-36) | 262 (50-574) | 10 (2-16) | 1.3 |
| tp | 1451 | 32 (16-39) | 26 (15-36) | 167 (43-474) | 6 (2-14) | 23 |
| tss | 1027 | 34 (20-42) | 33 (18-42) | 237 (123-500) | 7 (4-14) | 15.8 |
| wtemp | 1155 | 35 (19-42) | 33 (18-42) | 305 (182-573) | 10 (6-15) | 0.7 |

Time series duration and mean observation frequency for all other variables was generally similar to the *chla* time series. The median period of record was 32 years for both *tp* and *tn*. Median sampling frequency was 6 and 10 observations per year for *tp* and *tn*, respectively. A small proportion (2% and 1.8%, respectively) of *tp* and *tn* time series were shorter than 15 years. For

*tp*, 84%, 57% and 9% of the time series were longer than 20, 30 and 40 years, respectively. For tn, 83%, 61% and 5% of the time series were longer than 20, 30 and 40 years, respectively. There were 444 stations with joint *chla*, N and P observations for over 15 years. Among these, 220 corresponded to river stations, 169 to estuary stations, and 55 to lake stations.

Across all timeseries, the median temporal coverage was 1986 to 2022 (Table 3 and Figure 2). Yet, OLIGOTREND featured early and long *chla* time series with 19 of them starting before 1970 and an average of 50 year-long timeseries, most of them

found in the Lake PCI dataset. Across all variables, the 2000s and 2010s are the decades with the highest coverage. The 2020s were not as covered as the 2010s were, likely indicating that databases are not systematically updated with the most recent observations.

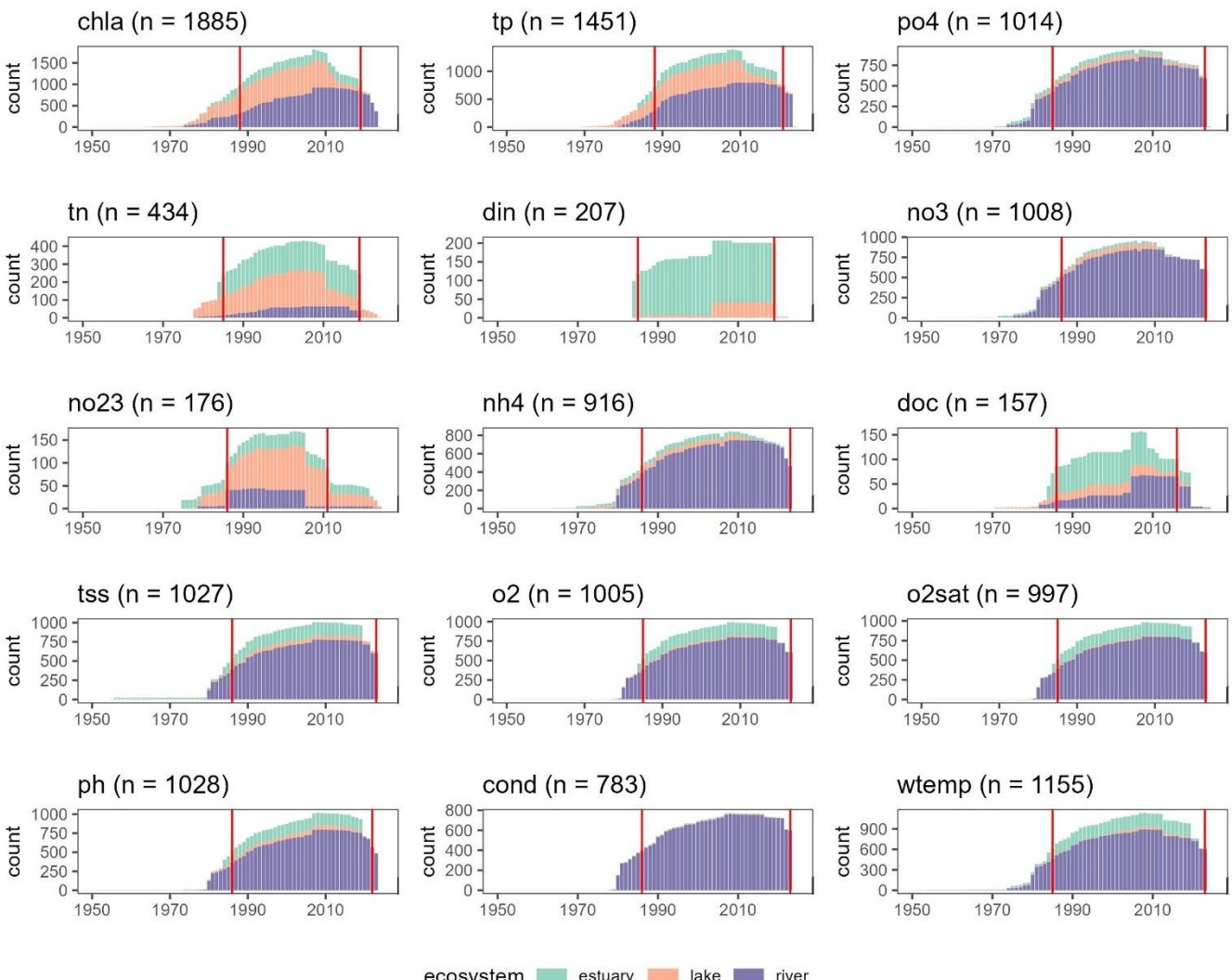

**Figure 2. Temporal coverage of OLIGOTREND timeseries for each environmental variable. The y-axis "count" shows the number of time series with valid observations for each year between 1960 and 2024. Only 35 time series started before 1960; 20 concerned *tss* and only one *chla*. Vertical red lines indicate median starting and ending years across the pooled dataset, i.e. the periods with the highest number of observations globally.**

### 3.2. Spatial coverage

The OLIGOTREND L1 database contains 13,992 time series originating from 1,894 sampling stations spanning across 5 continents (Table 1, Figure 3). There are 238, 687, and 969 stations located in estuaries or coastal bays, lakes or reservoirs, and rivers, respectively (Table 3). The 3 largest data sources are the French national water quality monitoring (775 stations), a global database of water quality measurements in lakes (Naderian et al., 2024 — 378 stations), and the United States' Chesapeake Bay Program (199 stations).

Geographically, the L1 dataset includes stations from 33 different large watersheds (Figure 3 and see Table S1 for detailed list

of these watersheds). The 5 most represented large watersheds are the Seine (France, 320 stations), the United States North Atlantic Coast (266 stations), the Mississippi-Missouri basin (231 stations), the French West Coast (183 stations), and England and Wales (163 stations). In total, 7 large watersheds contain more than 100 stations. Data from the Chesapeake Bay (United States North Atlantic Coast watershed) and the Elbe River watershed are particularly remarkable in terms of data contribution, covering hundreds of stations along the main rivers, encompassing both freshwater and estuarine zones.


**Table 3. Characteristics of the time series constituting the OLIGOTREND database, organized by data source (see Table 1). See Table S1 in the Supplementary material for similar statistics organized by basins. For the length of time series, number of observations per time series, and *chla* sampling frequencies, we provide the median value, and 10th and 90th percentiles are indicated in brackets.**

| Source | Median period of record | n stations (in estuary – lake – river) | Length [years] | $n_{obs}$ per time series | Average *chla* sampling frequency [n/year] | Total number of observations |
|---|---|---|---|---|---|---|
| naiades French water quality portal | 1988-2023 | 774 (24 - 1 - 749) | 34 (16-42) | 201 (71-416) | 4 (2-6) | 2,118,792 |
| Naderian2024 | 1986-2011 | 378 (0 - 378 - 0) | 25 (17-35) | 120 (37-260) | 6 (3-11) | 106,480 |
| Chesapeake Bay program | 1985-2019 | 199 (157 - 0 - 42) | 34 (19-35) | 408 (193-588) | 12 (10-17) | 822,961 |
| LAGOS-NE | 1985-2010 | 140 (0 - 140 - 0) | 24 (18-32) | 85 (35-248) | 5 (2-12) | 56,616 |
| UK Harmonized Monitoring Dataset | 1978-2012 | 133 (0 - 0 - 133) | 35 (20-44) | 299 (177-547) | 10 (6-15) | 168,474 |
| Lake PCI | 1988-2018 | 95 (0 - 95 - 0) | 23 (15-49) | 246 (116-1174) | 11 (5-21) | 93,580 |
| Danish monitoring program | 1983-2020 | 56 (0 - 56 - 0) | 33 (21-42) | 165 (33-481) | 6 (2-15) | 75,608 |
| Sacramento Bay Interagency monitoring | 1975-2021 | 46 (46 - 0 - 0) | 42 (18-46) | 297 (109-592) | 13 (7-18) | 50,126 |
| Elbe monitoring program | 1985-2016 | 25 (2 - 0 - 23) | 31 (22-38) | 581 (145-8490) | 15 (4-20) | 701,431 |
| Filazzola et al., 2020 | 2001-2018 | 13 (0 - 13 - 0) | 17 (16-28) | 123 (32-387) | 3 (1-12) | 7,852 |
| USGS-NWIS Data Retrieval | 1991-2021 | 10 (0 - 0 - 10) | 30 (21-31) | 682 (512-1093) | 22 (17-35) | 7,337 |
| GEMStat | 1980-2016 | 9 (0 - 3 - 6) | 26 (16-41) | 398 (158-645) | 11 (9-24) | 12,737 |
| LTER Florida Everglades | 1991-2008 | 9 (9 - 0 - 0) | 17 (16-33) | 207 (188-366) | 11 (10-12) | 25,027 |
| Danube River public program (HUN-REN CER, IAE) | 1979-2012 | 2 (0 - 0 - 2) | 33 (33-33) | 1100 (1010-1127) | 32 (32-32) | 13,032 |
| Victoria State Government | 1990-2024 | 2 (0 - 0 - 2) | 34 (26-34) | 782 (329-1685) | 39 (36-41) | 17,536 |
| Commission pour la Protection des Eaux du Léman (CIPEL) | 1980-2018 | 1 (0 - 1 - 0) | 38 | 815 (815-815) | 12 | 8,150 |
| Ebro River monitoring program | 1980-2004 | 1 (0 - 0 - 1) | 24 (15-24) | 284 (133-323) | 18 (18-18) | 2,039 |

| | | | | | |
|---|---|---|---|---|---|
| Romero et al., 2013 | 1982-2016 | 1 (0 - 0 - 1) | 34 (29-34) | 304 (176-362) | 4 (4-4) | 1,684 |
| TOTAL | 1986-2022 | 1,894 | 33 (17-42) | 220 (71-507) | 5 (3-14) | 4,281,312 |


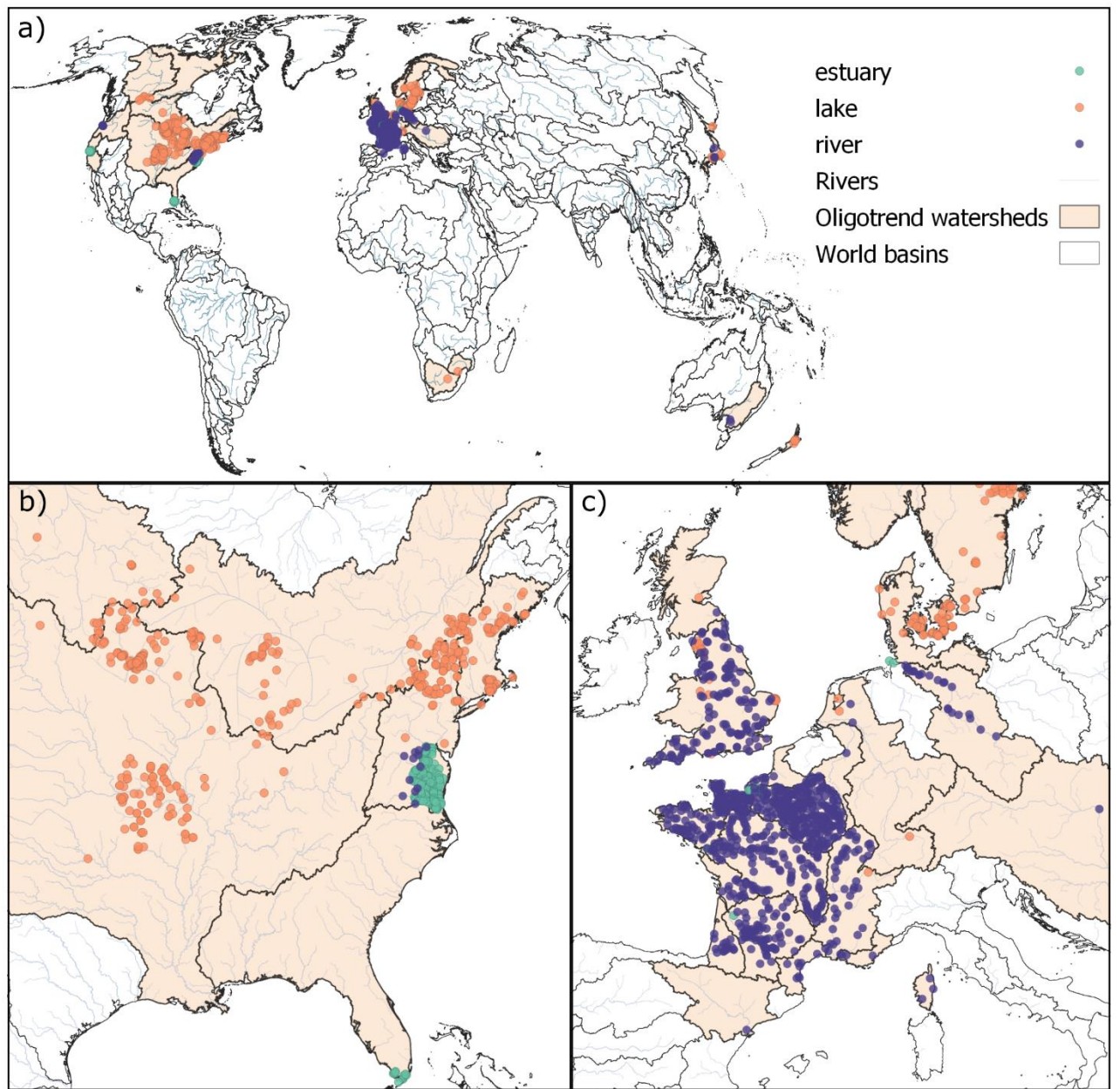

**Figure 3. a) Map highlighting the 1894 sampling stations included in the OLIGOTREND database at level L1, categorized by**

**ecosystem types. b) close-up on the Eastern side of the US, and c) on Europe showcasing most data points from France, UK and**
**Denmark.**

The OLIGOTREND database covers 1,229 sub-basins from the HydroATLAS database, distributing over 257 spatially independent large watersheds with no hydrological connections. OLIGOTREND covers a wide range of eco-physiographic contexts (Table 4). It covers medium to large watersheds (10th to 90th percentiles were 142 to 11,416 km$^2$), primarily lowlands. Stations extend to four climate zones, from extremely cold and mesic to hot and dry. Share among land-use types also covers
a wide range, from 100% forest or natural grassland areas to heavily impacted urban areas and croplands. Some of the stations are located in nearly pristine areas, but most of them are in highly populous areas.

Similarly, lakes and rivers represented by the OLIGOTREND database cover a wide range of morphometry, from shallow (e.g., Hickling Broad lake, England, average water column depth ~0.7 m) to deep and large lakes (e.g., Lake Geneva, France-Switzerland, average depth ~155 m), and from headwater streams (e.g., the Evel river in French Brittany draining a basin of
5 km$^2$) to large rivers (e.g., Mississippi, Danube, Rhine, Loire, Seine, Ebro, Susquehanna Rivers).

**Table 4. Basin characteristics covered by the OLIGOTREND database based on the HydroATLAS (level 12), the HydroLAKES and HydroRIVERS databases. Column "Range" indicates median values; and percentiles 10 and 90 are shown in brackets.**

| Category | Variable | Description | Aggregation | Range | Units |
|---|---|---|---|---|---|
| Physiography | up_area | Watershed area | Upstream sub-basin | 573.8 (142-11,416) | km$^2$ |
| | ele_mt_sav | Elevation | Sub-basin | 125 (28-417) | m a.s.l. |
| | slp_dg_uav | Terrain slope | Upstream sub-basin | 25 (10-71) | degrees |
| Climate | tmp_dc_syr | Air temperature average | Sub-basin | 10.1 (6.3-12.5) | degrees Celsius |
| | pre_mm_sy | Precipitation average | Sub-basin | 755 (625-1,106.2) | mm |
| | clz_cl_smj | Climate zone[(*)] | Sub-basin | 10 (7-11) | class |
| Land cover | for_pc_use | Forest cover extent | Upstream sub-basin | 15 (0-90) | % |
| | crp_pc_use | Cropland cover extent | Upstream sub-basin | 33 (4-64) | % |
| | pst_pc_use | Pasture cover extent | Upstream sub-basin | 10 (1-36) | % |
| Hydrology | dis_m3_pyr | Natural discharge | Sub-basin | 7.7 (1.5-131) | m$^3$/s |
| | run_mm_sy | Land surface runoff | Sub-basin | 376 (204-776) | mm |
| | lka_pc_use | Limnicity | Upstream sub-basin | 2 (0-60) | % |
| | dor_pc_pva | Degree of regulation | Upstream sub-basin | 0 (0-176) | % |
| Anthropogenic | pop_ct_usu | Population | Upstream sub-basin | 38 (2.5-874) | inhab. (x1000) |
| | ppd_pk_ua | Population density | Upstream sub-basin | 53.7 (11-294) | inhab./km$^2$ |
| | urb_pc_use | Urban cover extent | Upstream sub-basin | 2 (0-15) | % |
| Lake characteristics | Lake_area | Lake area | Lake body | 1.1 (0.2-25) | km$^2$ |
| | Depth_avg | Average lake depth | Lake body | 5 (2.9-14.7) | m |
| | Res_time | Residence time | Lake body | 289 (33-1394) | days |
| River characteristics | upland_skm | Watershed area | Upstream river segment | 629 (65-13,249) | km$^2$ |
| | dis_av_cms | Average interannual discharge | River segment pourpoint | 8.3 (0.8-143) | m$^3$/s |

| | | | | |
|---|---|---|---|---|
| ord_stra | Strahler order | River segment | 3 (2-5) | d.l. |

*: Climate zone classes encompass the following classes: Extremely cold and mesic, Cool temperate, Warm temperate and Hot and dry.

### 3.3. OLIGOTREND time series ranges and relationships

For most variables, long-term averages are clustered by ecosystem type (Figure 4). The lowest *chla* concentrations were found in rivers (7.8 ± 10.7 ug L$^{-1}$) followed by estuaries (11.8 ± 9.9 ug L$^{-1}$) and then lakes (18.0 ± 25.3 ug L$^{-1}$). This greatly contrasted with most P, N, and oxygen time series: for instance, *tp* and *tn* distributions showed the highest ranges in rivers (0.13 ± 0.11 mg P L$^{-1}$ and 3.1 ± 1.8 mg N L$^{-1}$), and the lowest in lakes (0.06 ± 0.13 mg P L$^{-1}$ and 1.9 ± 0.9 mg N L$^{-1}$). For DOC, most time series remained within a similar range of values regardless of ecosystem type, except for four lakes located in the North-East

US (global lake database; Naderian et al., 2024). The highest conductivity values appeared in estuaries, much higher than in rivers or lakes. There were only 9 lakes with conductivity time series, explaining the density distribution peaks for this ecosystem type. The warmest waters were also found in estuaries.

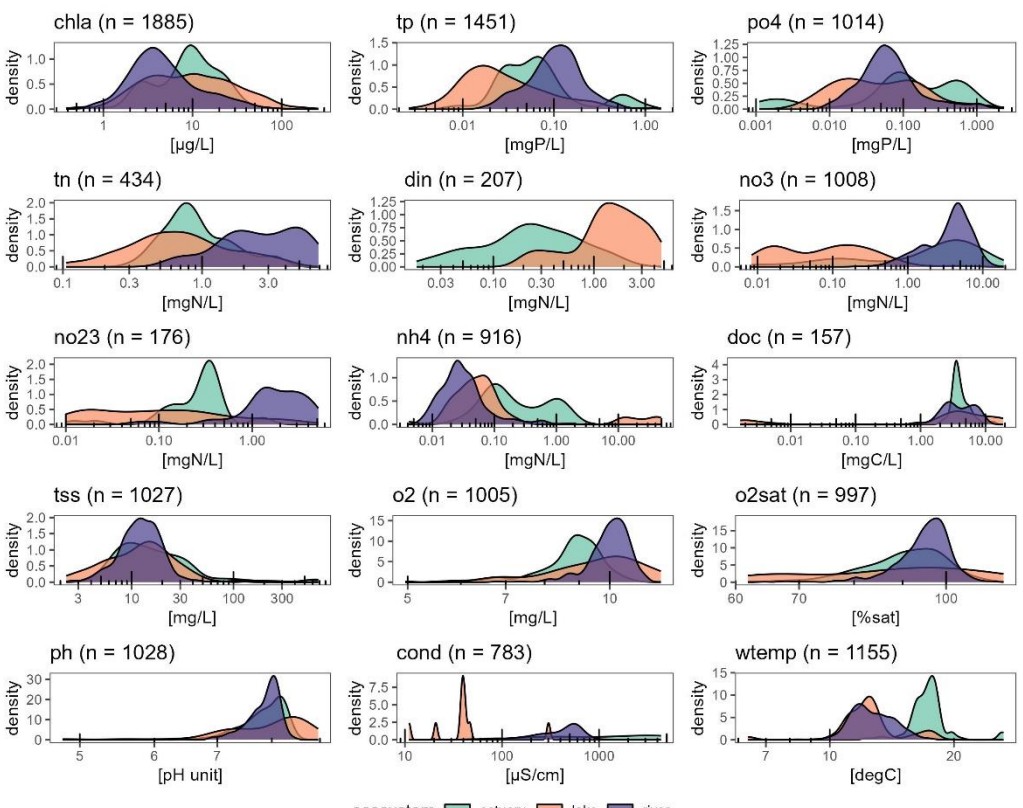

**Figure 4. Distribution of inter-annual average concentrations of all the OLIGOTREND time series. Number of time series for each**
**variable are indicated in brackets for each variable.**

Across the entire database, *chla* annual averages showed moderate to strong correlation with *tp* and *tn* (Figure 5). *Chla* was strongly and positively correlated with *tp* (Pearson, r = 0.39) across all ecosystem types. The positive correlation was the

strongest for lakes (r = 0.82), moderate for rivers (r = 0.37), and estuaries showed the weakest relationship (r = 0.29). *Chla* was positively correlated with *tn* (Pearson, r = 0.40), which was the highest in lakes (r = 0.75), moderate in rivers (r = 0.49)

and lowest in estuaries (r = 0.30). Variables *tp* and *tn* were positively correlated across all ecosystem types (Pearson, r = 0.59), with the strongest correlation found in lakes (r = 0.74), slightly lower in rivers (r = 0.63), and the weakest one in estuaries (r = 0.35). There was a clear cluster outlier for these variables in estuaries, characterized by low *chla* and *tp* but rather high *tn*. These observations corresponded exclusively to the Florida Coastal Everglades.

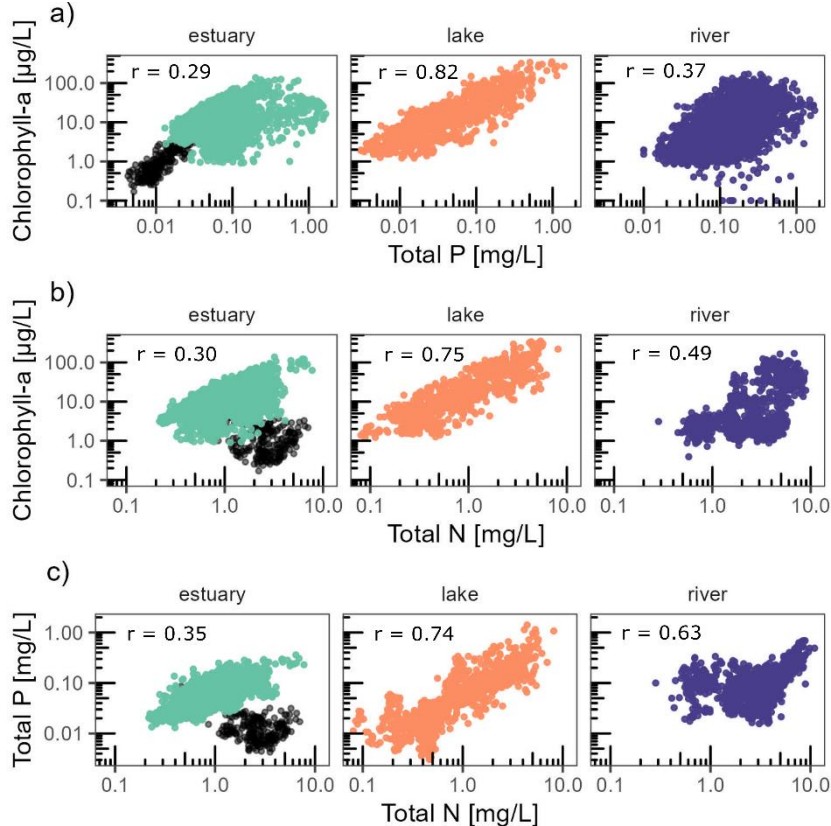

**Figure 5. Relationships between *chla* and *tp* (a), *chla* and *tn* (b), and *tp* and *tn* (c). Each dot represents the annual mean for a given time series. Dark dots for estuary stations highlight the observations in the Florida Coastal Everglades which clearly stand out from all other estuarine observations. Pearson correlations are all statistically significant (p-value < 2e-16) and corresponding coefficients (r) are indicated in each panel.**

### 3.4. Trends in the OLIGOTREND database

Comparing the mean value of annual averages between the second and the first halves of time series proved to be a simple but effective way to overview temporal behaviour of time series in the database. Across all variables and ecosystem types, 60% of time series showed a lower average value in the second half. 63% of *chla* time series showed lower values in the second half (Figure 6). For N and P nutrient time series, 78% to 87% showed an average concentration lower in the second half (it was 85%, 87%, 78%, 85%, 86% for *tp*, *po4*, *tn*, *din*, *nh4*, respectively). An exception was found for *no3* with only 45% time

series with a lower concentration in the second half of the time series. Interestingly, we found that the majority (74%) of *tss* time series had a lower concentration in the second half, whereas *o2*, *o2sat*, *pH*, and *cond* showed no clear differences in the second half of the time series with 49%, 43%, 42%, and 42%. For *wtemp*, there was a clear indication of a warming trend with 64% of time series with higher averages in the second half of the time series.

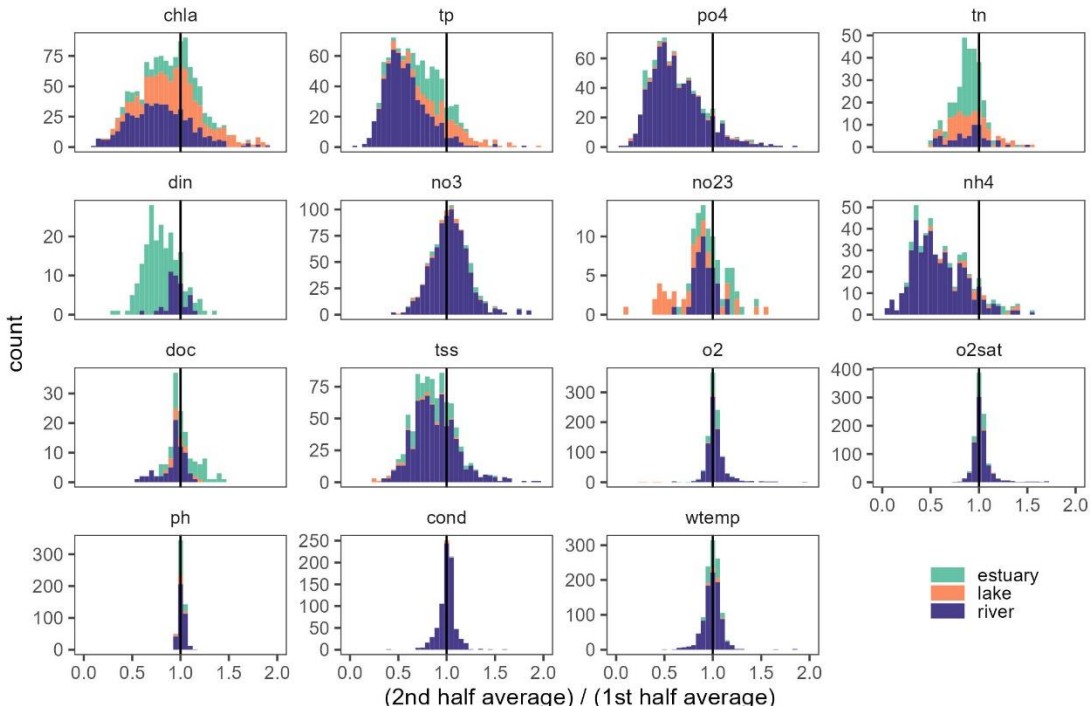

**Figure 6. Distribution of ratio between 2nd half time series averages over 1st half averages. Values significatively below 1 likely indicate declining trends, regardless of the complexity of the temporal trajectory.**

The breakpoint and trend analysis (Figure 7) revealed 15% of *chla* time series were best represented with a segmented trend component while 62% had no trend detected, 18% presented a monotonic declining trend, 5% a monotonic rising trend (predominantly found in estuaries, see Figure 8). The predominant segmented trend types were "00" (32%), "0-" (21%), "-0" (19%) and "+-" (7%).

For *tp* and *po4*, 29-31% of the time series had a breakpoint with a segmented trend, 26-32% had no trend detected, while 35-42% presented a declining monotonic trend and 1-2% were rising. For *tp* time series, 72% of segmented trends had a declining trend type, while it was 65% for *po4* time series. Compared to rivers and estuaries, a lower proportion of declining *tp* trends were observed in lake time series.

For N species, time series were dominated by the no-trend type (38-61%) and significant trends were contrasted: *tn*, *din* and *nh4* showed a large number of declining trends (36-42%) and a small proportion of rising trends (less than 2%), while *no3* and *no23* were characterized by a larger proportion of rising trends (7% for *no23* and 17% for *no3*) and segmented trends (14% for *no23* and 25% for *no3*). For *no3*, 57% of segmented trends had a declining trend type on the most recent part of the time

series as 34% were "0-" and 23% were "+-". Other variables were characterized by 50-60% of no-trend time series.
Interestingly, among the detected trends, *tss* showed a significant proportion of declining trend types, while *o2*, *o2sat*, *pH* and *wtemp* showed a predominance of rising trends.

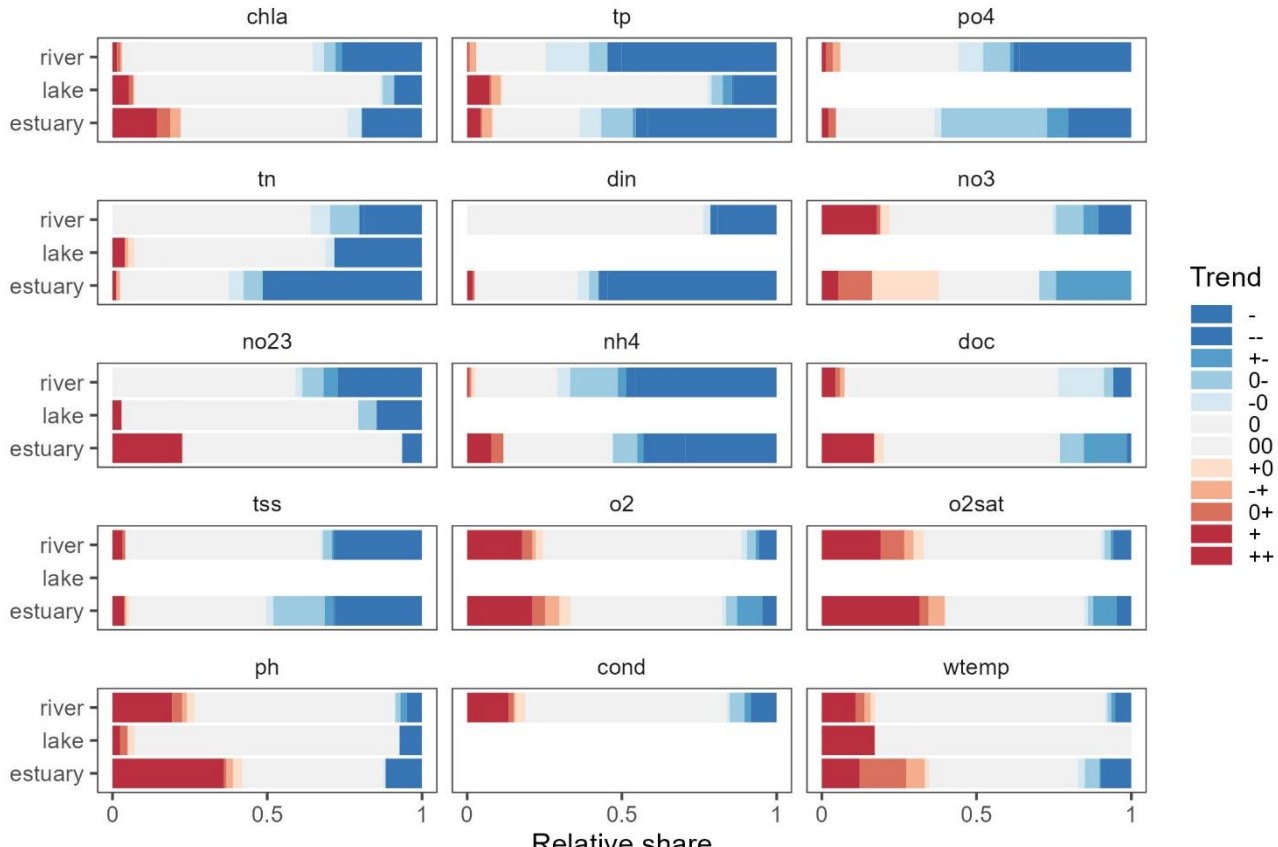

**Figure 7. Overview of trend significance and trend types identified in the OLIGOTREND database. Blue stripes are indicative of declining trends, grey stripes of no-trend, and red stripes of rising trends. Empty stripes indicate variables or ecosystems where the number of time series available was lower than 30. Refer to Section 2.4 for a detailed explanation of trend symbols indicated in the legend.**

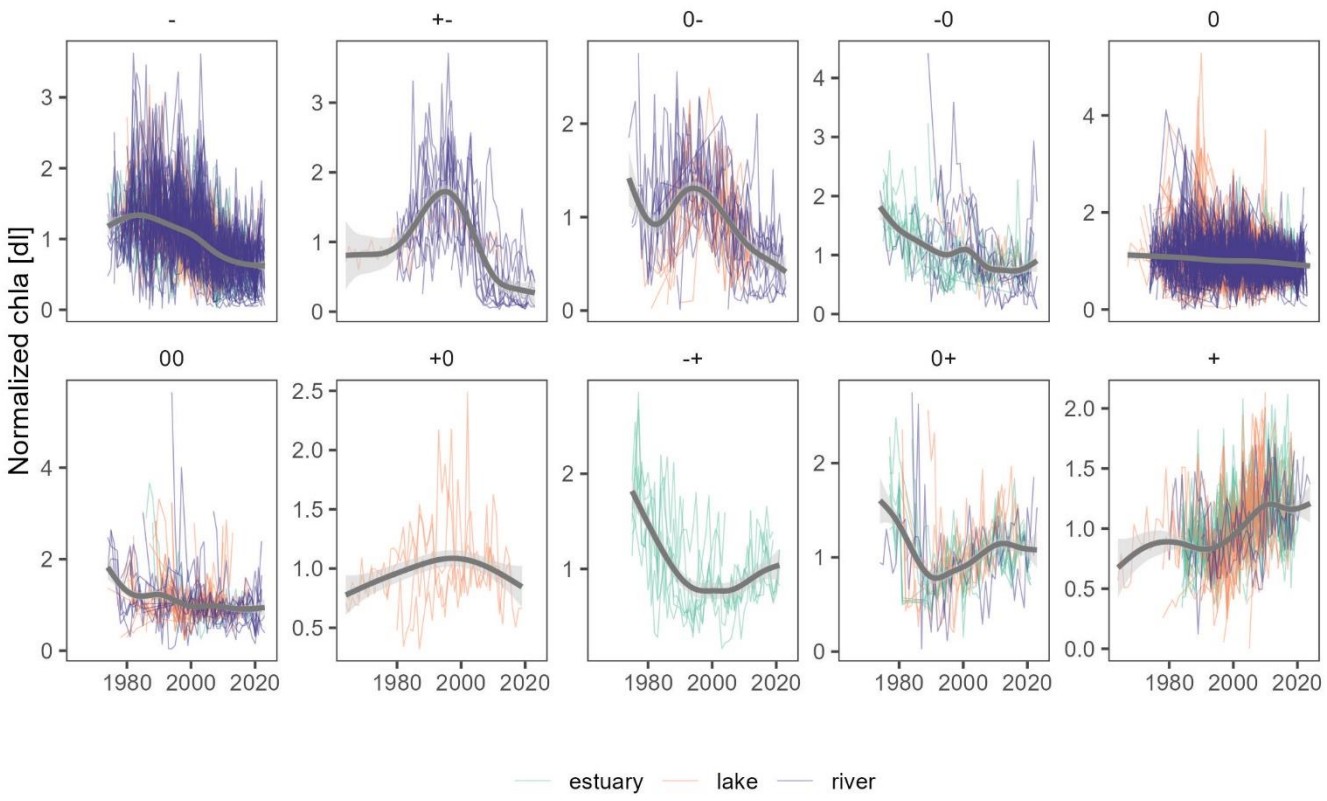

**Figure 8. Overview of all *chla* annual time series normalized by interannual averages (thin lines), organised by trend types (panels) and ecosystem type (colour). Thick grey lines are smoothed curves of all time series within a given panel, only displayed to guide the reader. Refer to Section 2.4 for a detailed explanation of trend symbols indicated on top of each panel.**

For *chla*, Sen's slopes in estuaries were smaller in magnitude compared to lakes and rivers, regardless of the trend type (Figure 9a). Lakes exhibited a median Sen's slope of –0.7 µg $L^{-1}$ year$^{-1}$; it was –0.4 µg $L^{-1}$ year$^{-1}$ in rivers and -0.3 µg $L^{-1}$ year$^{-1}$ in estuaries. The fastest declines (below –4 µg $L^{-1}$ year$^{-1}$) were found in the Sacramento Bay in California, the River Loire (France), and several shallow lakes in the Mississippi-Missouri basin, the Denmark Germany Coast, and England and Wales. The largest positive *chla* trends were found in rivers, with a median slope of 0.79 µg $L^{-1}$ year$^{-1}$, compared to 0.13 and 0.23 µg $L^{-1}$ year$^{-1}$ in estuaries and lakes, respectively. The fastest rises (above 4 µg $L^{-1}$ year$^{-1}$) were found in the River Loire (France). For *tp*, the fastest rises and declines were observed in river ecosystems (Figure 9b) with median slopes of 4.0 x10$^{-3}$ and –4.7 x10$^{-3}$ mgP $L^{-1}$ year$^{-1}$, respectively, one order of magnitude greater than the slopes observed in lakes and estuaries. The fastest declines (below –0.1 mgP $L^{-1}$ year$^{-1}$) were observed in the Rhône and Seine Rivers (France).

For *tn*, although the fastest declines were observed in estuary stations (Florida Coastal Everglades) down to –0.4 mgN $L^{-1}$ year$^{-1}$, the median value for declining slopes was overall faster in rivers with median slopes of –0.14 mgN $L^{-1}$ year$^{-1}$ (Figure 9c). It was –6 x10$^{-3}$ mgN $L^{-1}$ year$^{-1}$ in estuaries and -7 x10$^{-3}$ mgN $L^{-1}$ year$^{-1}$ in lakes. Only 11 stations showed rising *tn* trends (Figure 7), and among them, 3 were in the Chesapeake Bay (US North Atlantic Coast) which contrasted with the 145 other

estuarine stations in this basin which either showed declining trends (n=89) or no trends (n=56). Note that only 7 lacustrine stations showed rising *tn* and in rivers, and none of the *tn* time series showed a rising pattern.

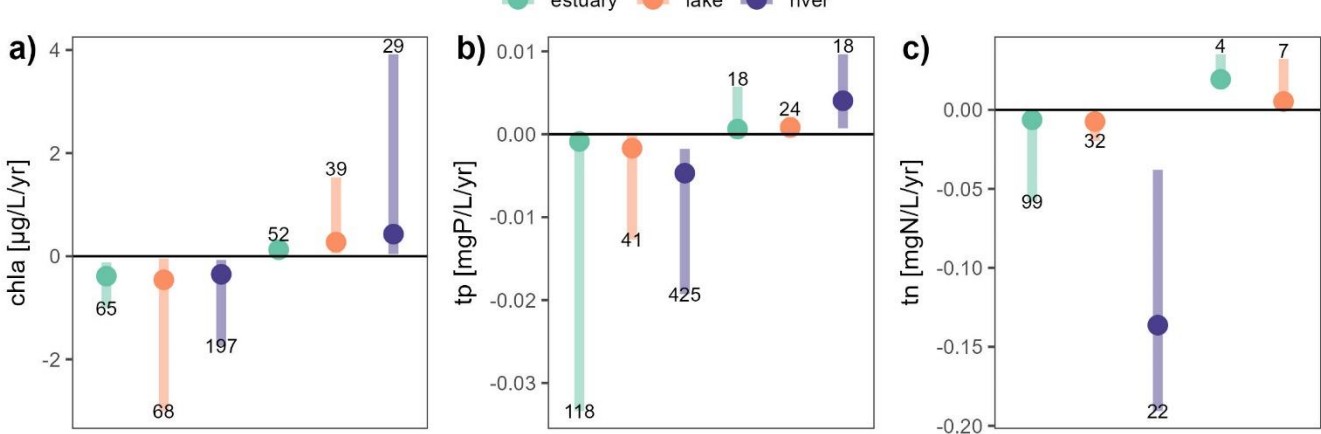

**Figure 9. Overview of all Sen's slopes calculated for *chla* (a), *tp* (b) and *tn* (c) whether they are showing a declining (negative values) or a rising trend (positive values). Medians by ecosystem type are indicated with a plain circle, 10th and 90th percentiles correspond**
**to the segment ends. The numbers of time series found for each variable, ecosystem and trend type are indicated at the bottom or the top of each segment. See Fig. S1 in the Supplementary material for a similar figure for all variables included in OLIGOTREND.**

We identified 444 stations with joint *chla*, P and N data over 15 years and more than 6 observations per year. Among these, 100 (or 23%) *chla* time series showed a linear declining trend, 251 (or 57%) had no trend, and 37 (or 8%) were rising. Declining *chla* time series were also linked to declining trends in N and P (Figure 10a). Nearly half of the *chla* time series with no trend 355    had corresponding no-trend or declining patterns in nutrient time series (Figure 10b). Rising *chla* time series predominantly corresponded to no-trend or declining patterns in nutrient time series. Only 18% of the rising *chla* time series also had significant rising trends in N or P.

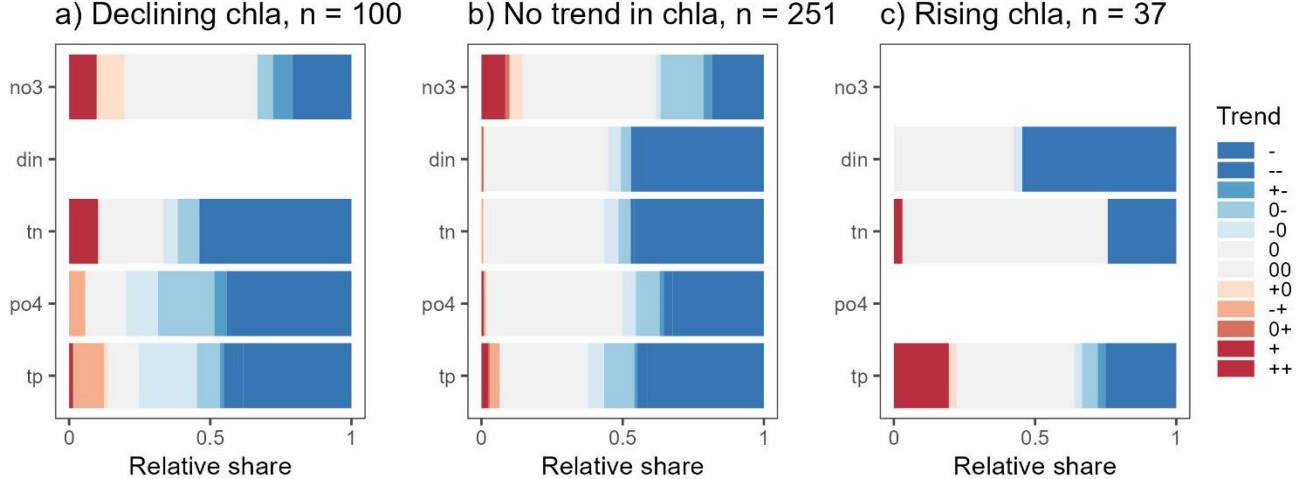

**Figure 10. Relative share of trend types found for nitrogen and phosphorus concentrations related to *chla* time series with declining**

 **trends (a), no trends (b), and rising trends (c). This analysis is based on 444 stations having parallel measurements of *chla*, and N (*din* and/or *no3* and/or *tn*) and P (*po4* and/or *tp*) for at least 15 years. Empty rows correspond to variables with less than 30 time series.**

## 4. Potential implications of OLIGOTREND for future research

The OLIGOTREND database has the potential to answer some important questions in large-scale aquatic ecology, biogeochemistry, and global change studies. Below, we highlight the most important findings of the database and discuss potential implications for future research beyond disciplinary boundaries.

### 4.1. Unravelling the ambiguous links between *chla* and nutrient levels for lakes, rivers, and estuaries

The development of primary producers is far more complex than a single relationship with nutrient availability, especially if one also considers the differences among ecosystem types. Hydraulic flushing, turbulence, exposition to solar radiation, temperature (e.g. Reynolds, 2006), and light climate (Hilt et al., 2011) are crucial environmental variables in lotic systems. Water residence time, internal loading (Jeppesen et al., 2005; Krishna et al., 2021), stratification regime, and underwater light climate are other crucial factors controlling lentic ecosystems (Donis et al., 2021). Such differences are also reflected in the OLIGOTREND database. For instance, on one hand rivers had the highest P and N concentrations, followed by estuaries and lakes, and on the other hand, the highest *chla* concentrations were found in lakes followed by estuaries and then rivers (Figure 4). Further, only 18% of the *chla* time series showed a linear declining trend which contrasted greatly with a dominating decreasing trend for most nutrient concentrations (Figures 6, 7 and 10). Moreover, although lake time series showed the highest correlation between *chla* and nutrients (Figure 4), they were also the ones with the highest proportion of non-significant trends (Figure 7). In this context, we argue that the OLIGOTREND database provides a unique opportunity and foundation to further investigate the ambiguous links existing between *chla* and nutrient levels over many contrasted water bodies located in basins with different environmental and climatic conditions.

### 4.2. Is oligotrophication specific to aquatic ecosystem types?

The OLIGOTREND database evidenced different responses of the individual ecosystem types to nutrient declines (Figures 7, 8 and 10). For instance, compared to estuaries and lakes, rivers showed the highest proportion of declining *chla* (Figure 7). The inherent specificities of different ecosystems could partly explain why oligotrophication seems to be ecosystem-specific: i) the successful P reduction in many rivers worldwide (e.g., Le Moal et al., 2019) has led to more frequent P limitation for phytoplankton (Elser et al., 2007), although N or Si may also be limiting primary production (Paerl et al., 2016); ii) in lakes, longer water residence time, and internal nutrient loading can either delay (Jeppesen et al., 2005) or amplify (i.e., through algal blooms; e.g., Krishna et al., 2021) the ecological response following nutrient declines; iii) temporal shifts in phytoplankton assemblages towards taxa better adapted to low P levels, or taxa that are barely controlled by zooplankton grazing (e.g. filamentous cyanobacteria; Selmeczy et al., 2019) can often represent overlooked effects explaining rising or weak trends in

primary producers despite nutrient decline over time (Anneville et al., 2019); iv) in estuaries, the dynamic of primary producers is also largely affected by marine waters, where coastal phytoplankton, sensitive to N (Elser et al., 2007), or N and P availability meets freshwater phytoplankton primarily sensitive to P (Kemp et al., 2005). Future analysis of OLIGOTREND time series together with catchment and waterbody attributes could improve our understanding of how aquatic ecosystems respond to
nutrient trends in a wide variety of aquatic ecosystems.

### 4.3. Abrupt and gradual changes in long-term water quality time series

The OLIGOTREND database could be explored to further evidence the extent of gradual changes or abrupt regime shifts in water quality time series. In fact, some of the waterbodies represented in OLIGOTREND are known for shifting their primary producer's structure and function following oligotrophication. This is the case of the Loire (France) and the Ebro Rivers
(Spain), which are known for their long-term gradual regime shifts from phytoplankton to macrophytes in response to phosphorus decline (Diamond et al., 2021; Ibáñez et al., 2012; Minaudo et al., 2015, 2021). Similarly, phytoplankton of the middle Danube now more frequently contains benthic taxa, predominantly diatoms, potentially indicating a long-term regime shift from pelagic to benthic production in recent decades (Abonyi et al., 2018). Moreover, oligotrophication can result in a shift from heterotrophic conditions to dominantly autotrophic processes with lower pollution, as observed for the Elbe River
(Wachholz et al. 2024). OLIGOTREND time series could be further analysed to detect possible temporal changes in variance (as a possible early-warning signal, Dakos et al., 2015), seasonal patterns, and relationships between *chla*, nutrients and ecosystem metabolism. This could enhance our understanding of crucial factors underlying regime shifts in river ecosystems, which are comparatively less well known than in lakes (Gilarranz et al., 2022).

In OLIGOTREND, we highlighted a significant number of no-trend or rising *chla* time series despite declining nutrient levels
(Figure 10c). This could be related to climatic effects and long-term changes of ecosystem structure, such as in the Chesapeake Bay (Harding et al., 2019). Future analysis of the OLIGOTREND will provide an invaluable source of data to disentangle the effects of climate change and watershed biogeochemistry on multi-decadal *chla* and nutrient trends.

### 4.4. Combining OLIGOTREND with large-scale datasets to foster interdisciplinary aquatic data science

The OLIGOTREND database can help boost water quality research if it is combined with other large-scale or long-term
ecological datasets. For instance, it is known that shifting baselines because of temporal changes in different, covarying environmental factors can preclude the return of primary producer to pre-eutrophication conditions (Carstensen et al., 2011; Duarte et al., 2009). As global change intensifies, leading to novel ecosystems (Hobbs et al., 2009), the temporal extension of most available water quality datasets limits a correct estimation of pre-eutrophication baselines. Only a fraction of the OLIGOTREND database covers *chla* and/or nutrients during the eutrophication phase, which renders pre-oligotrophication
reference conditions impossible to discern; and hence, makes it difficult to validate nutrient remediation actions (Pinay et al., 2017). In this context, combining paleolimnological observations with water quality monitoring data could have a potential

not fully implemented at large spatial scales and across different aquatic ecosystem types (Bennion et al., 2015; Bhattacharya et al., 2022; Dong et al., 2012).

Recent research has shown that nutrient concentrations link to nutrient loads (point and nonpoint sources) at the catchment scale (Ehrhardt et al., 2021; Jarvie et al., 2012; Murphy et al., 2022). Yet, only a few studies have established a mechanistic link between nutrient input management and the development of the phytoplankton biomass. Data-based approaches that jointly analyse decreasing nutrient loadings over multiple decades and sites with corresponding measurements of *chla* and nutrients can help better characterize how successful catchment management and environmental measures can be to reverse eutrophication. OLIGOTREND holds the potential to approach oligotrophication longitudinally at the basin scale, where long-term trajectories can be assessed from small streams, rivers, lakes/reservoirs towards estuaries/coastal ecosystems along with their hydrologically connected time series.

Remote sensing could further supplement crucial water quality information organised in OLIGOTREND. Remote sensing can provide time series data on water quality for inland and coastal aquatic ecosystems, which, if combined with in-situ measurements, can increase *chla* data coverage both spatially and temporally (Ross et al., 2019; Spaulding et al., 2024). Moreover, regional and Earth System numerical models will improve further if calibrated or validated by in situ observations (Casquin et al., 2024; Liu et al., 2024). The OLIGOTREND database readily represents a centralized and harmonized dataset open for calibration and validation by remotely sensed water quality data, and available for training and validating regional and large-scale numerical models.

Finally, there is a growing interest in large-scale observations that integrate new and existing databases to answer key questions in aquatic ecology (Barquín et al., 2015). Long-term observations of community data (e.g. via LTER and eLTER, GBIF, Biofresh) may include key functional groups of aquatic food webs, such as phytoplankton, zooplankton, macroinvertebrates (Welti et al., 2024), and fish (Comte et al., 2021). For a selection of sites, *chla* trends can be further analysed jointly with long-term community data to investigate the role that community composition and biodiversity may play in responding to long-term environmental change (Jochimsen et al., 2013). Some of the OLIGOTREND time series are linked to lotic community data (i.e., phytoplankton), which have been seldom explored compared to lakes when testing the biodiversity effect on ecosystem functioning and services (Filstrup et al., 2019; Ptacnik et al., 2008).

## 5. Code and data availability

All the data are openly available along with the R scripts used for data processing from raw measurements at L0a level to higher data processing levels. All R scripts produced to extract, harmonize and process the OLIGOTREND data were stored and organized in a dedicated GitLab repository (https://gitlab.com/OLIGOTREND/wp1-unify). Data at levels L1 and L2 (Figure 1) were deposited in an Environmental Data Initiative Data Package accessible on the EDI data portal (https://doi.org/10.6073/pasta/a7ad060a4dbc4e7dfcb763a794506524, Minaudo & Benito, 2024). Original links to data sources of L0a data are provided in Table 1 and in the EDI Data Package. Additionally, we also provide in the GitLab repository all

the GIS files emerging from the data extraction step, including shapefiles of L0 and L1 stations, the corresponding basins,
lakes and rivers characteristics resulting from the spatial join between OLIGOTREND stations and the HydroATLAS.

## 6. Conclusions

The OLIGOTREND database provides invaluable information in aquatic ecology and Earth system science. We evidenced oligotrophication at large temporal and spatial scales and unveiled the complexity of the chlorophyll-*a* response following oligotrophication and the relationships between chlorophyll-*a* and nutrients in inland and transitional waters covering a wide
range of climatic and environmental conditions. While the database is not exhaustive, its flexible structure and reproducible processing pipeline facilitate the inclusion of additional datasets in the future. We also see a strong need to continuously update the database due to the accelerating climate change and the resulting impacts on the loading and processing of nutrients and the associated ecological implications (van Vliet et al., 2023). Finally, OLIGOTREND will support collaborative efforts aimed at advancing our understanding of the complex biogeochemical and biological mechanisms driving oligotrophication and the
broader ecological impacts of global environmental change.

**Author contributions**

CM and XB both secured the funding for this study and contributed equally for the Conceptualization, Methodology, Data curation, Formal analysis, Investigation. They wrote together the original draft. All other authors provided datasets and participated in the revisions of the original draft.

**Competing interests**

The authors declare that they have no conflict of interest.

**Funding sources**

This study was funded by the Iberian Association of Ecology (SIBECOL) through the 2022 early-career advanced grant to CM and XB. Both CM and XB have received funding from the postdoctoral fellowships programme Beatriu de Pinós, funded
by the Secretary of Universities and Research (Government of Catalonia) and by the Horizon 2020 programme of research and innovation of the European Union under the Marie Sklodoska-Curie grant agreement No 801370. AA was supported by the National Research, Development and Innovation Office, Hungary (FK 142485 project) and by the János Bolyai Research Scholarship of the Hungarian Academy of Sciences. ER acknowledges the support of the Grant "Severo Ochoa Centres of Excellence" (CEX2018-000828-S) funded by MCIN/AEI/10.13039/50110001103, and the project KALORET (PID2021-
128778OA-I00) funded by MCIN/AEI/ 10.13039/501100011033 and by "ERDF A way of making Europe".

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
