# Peer review of "OLIGOTREND, a global database of multi-decadal chlorophyll-a and water quality time series for rivers, lakes and estuaries"

_Earth System Science Data, 2025_

## Author Response (AR2)

**Response to Referee 1**

**General comments**

Referee1 Comment 1 (R1C1): The authors highlight the need for a unified dataset that links chlorophyll-a (chla) and nutrient measurements in freshwater as the main motivation behind OLIGOTREND. The resulting dataset focuses on high-income countries, meaning gaps outside North America and Western Europe remain. Still, OLIGOTREND successfully improves the current spatial and temporal coverage of the aforementioned water quality (WQ) parameters, making it applicable for large-scale oligotrophication studies. I found the manuscript structure logical and the writing concise and up to ESSD standards. I particularly appreciate the clarity of the data processing workflow and the inclusion of complementary GIS layers.

Below is a list of my remarks and suggestions, mainly minor technical corrections related to wording. Finally, I recommend a minor revision before the final acceptance of the manuscript in ESSD.

Authors: We are very grateful for the positive appreciation and all the suggestions and comments provided by the Referee. Below, we answer point-by-point to each comment and issue raised by the Referee.

**Specific comments**

R1C2: You have listed the summary statistics of the dataset in the abstract (L32-33). I suggest adding the overall temporal coverage of the dataset here, too. Although the time series length depends on the parameter, you can use the earliest and latest years in OLIGOTREND to give a general estimate for users.

Authors: We agree with this suggestion. The earliest and latest years in OLIGOTREND correspond to 1933 and 2024, respectively. However, only one lake in the Mississippi-Missouri basin presents such an exceptionally long period of record. Instead, we identify that the majority of OLIGOTREND timeseries cover the period 1986 to 2022 and this is information added in the revised version of the manuscript as requested as follows:
L35-36: *"Most time series covered the period 1986-2022 and comprised at least 15 years of chlorophyll-a measurements."*

R1C3: I appreciate the well-defined data processing workflow shown in Fig. 1 (section 2). I particularly like that you used "levels" to label the different modification stages of the dataset, e.g. "L0a" for raw data. These make it easier to follow the workflow as it moves through the processing pipeline.

Authors: Thank you!

R1C4: I commend the authors for providing catchment boundaries and attributes along with the WQ measurements (section 2.3), which enhances the applicability of the data for modelling purposes. However, due to coarse resolution, the boundaries of HydroATLAS layers are often quite inaccurate compared to national-level datasets. Did you consider any alternative sources by any chance, e.g. the CAMELS catchments?

Authors: We acknowledge catchment boundaries extracted from HydroATLAS can be sometimes inaccurate compared to national-level datasets.

CAMELS catchments are a fantastic resource, but they concern sites located only in the US, making it difficult to find similar resources elsewhere. This is why we prefer linking OLIGOTREND sampling sites to a uniform and global database. In any case, users can easily access the coordinates of all sampling sites and join OLIGOTREND with any other spatial database that they find suitable for their own purposes.

R1C5: You mention providing the GIS data emerging from the data extraction step (L423-425). I understand the GIS data is currently available in the GitLab repository rather than the EDI data portal. I cloned the GitLab repository and accessed the GIS layers, so this was not an issue for me. However, I suggest providing at least some of the GIS data (e.g., catchment boundaries) together with the WQ data in the EDI portal. This would make GIS data access more convenient for the non-technical user.

Authors: This is a great suggestion, although users can already access and download the csv table "oligotrend_L1_xy_gis.csv" with the geographic coordinates of all OLIGOTREND sampling sites from the EDI portal.

We prefer to keep these GIS layers accessible only in the GitLab repository for the following reasons:

- Catchment boundaries can only be found for lake and river stations, excluding estuarine/coastal sampling sites.
- A fraction of lake and river stations could not be successfully linked to any HydroSHEDS catchment, and we would rather make sure users, including non-technical users, are fully aware of it rather than using blindly the catchment boundaries they would have downloaded from the EDI data portal.
- Anyone willing to download all the GIS layers can access them directly in the GitLab repository without needing to clone the entire OLIGOTREND repository.

R1C6: You plan to continuously update OLIGOTREND in the future (L431-433) in the conclusions. Considering that the dataset mostly consists of measurements from high-income countries, resulting in gaps globally, would these updates also include improving the spatial coverage? For example, would leveraging remote sensing be an option to fill some of these gaps?

Authors: In the future, OLIGOTREND can be improved with a larger spatial coverage and by updating existing timeseries with more recent observations. Remote sensing could be an option to improve spatial coverage of chlorophyll-a observations in a much larger number of waterbodies. However, this would come with the disadvantage of

introducing a very different type of observation, with rather large uncertainty in chlorophyll-a estimates, and possibly impacting significantly the quality of trend analyses. If one were to include this type of observation, we'd recommend explicitly treating chlorophyll-a observations, whether they are derived from *in situ* measurements or from Earth Observations.

As it is now, we think OLIGOTREND has already an outstanding potential to improve our understanding of the temporal and spatial patterns of chlorophyll-a across aquatic ecosystems. Finally, the strength of OLIGOTREND may reside in the possibility of exploring the relationship between long-term chlorophyll-a, community data if available beyond chlorophyll-a, and other water quality variables across a broad range of eco-biogeographical contexts.

**Technical corrections**

Throughout the manuscript: "timeseries" -> "time series"

L53: "understanding of oligotrophication is still not fully understood" -> Perhaps "our understanding is incomplete" to avoid repetitiveness.

L66: "across-ecosystem" -> "cross-ecosystem"

L80: "geo-spatial" -> "geospatial"

L109: "Kjeldhal" -> "Kjeldahl"

L128: "was" -> "were"

L132-133: "To offer the possibility to OLIGOTREND users to design their own quality check procedure, we did not remove any data in response to data curation (QA/QC)." -> "We did not remove any data in response to data curation (QA/QC) to allow users to design their own quality check procedure."

L152: Please add the QGIS version.

L195: "Table 2. Overview of La data" -> Should it be "L1 data"?

L353: "nutrients" -> "nutrient"

L396: "inputs" -> "input"

Please unify the mixed use of British and American spelling in verbs like "analyse" (L397) and "characterize" (L398) throughout the manuscript.

Authors: Thank you for highlighting all these points; they have all been addressed in the revised version of the manuscript.

**Response to RC2**

Referee2 Comment 1 (R2C0): This is an important and very well-written paper, presenting an important dataset with broad applicability across environmental and ecological fields. Figures are clear and very informative. Despite its spatial bias, OLIGOTREND offers a solid foundation for understanding patterns of oligotrophication at large scales for different water body types. I also commend the authors for the inclusion of a flexible and reproducible processing workflow, which will facilitate the continued expansion of the database in the future. OLIGOTREND will also help to support the development of analyses that can inform water quality management and monitoring efforts across diverse regions.

Authors: We are very grateful for the positive appreciation and all the suggestions and comments provided by the Referee. Below, we answer point-by-point to each comment and issue raised by the Referee.

I think the authors should consider three main points that require further clarification:

R2C1.   It was unclear to me what was the temporal coverage of the datasets for different water quality parameters. Can you add a panel in figure 2 showing a violin plot or something similar that depicts the distribution of datasets across years per water quality parameter and classified by water body type?

Authors: We realize this was not clearly shown in the manuscript. Although Table 3 shows the median temporal coverage across all variables and by data source, we have produced an additional figure (now Figure 2) to provide the reader with more detailed information on the temporal coverage of OLIGOTREND for all variables included.

In this Figure, we count the number of times the OLIGOTREND time series covers a given year, from the earliest measurements (1960) to the latest (2024).

[Figure]

**Figure 2. Temporal coverage of OLIGOTREND timeseries for each environmental variable. The y-axis "count" shows the number of time series with valid observations for each year between 1960 and 2024. Only 35 time series started before 1960; 20 concerned tss and only one chla. Vertical red lines indicate median starting and ending years across the pooled data set, i.e. the periods with the highest number of observations globally.**

R2C2. I think there is a high risk of matching the monitoring stations with some attributes from HydroATLAS, as there may be substantial mismatches between the temporal coverage and the spatial scale of each water quality dataset and layers such as land cover data. Wouldn't it be better to link characteristics from national datasets when possible, rather than relying solely on HydroATLAS? Or at least you should add a flag indicating if the trend data aligns with the temporal scale of the land cover characteristics and anthropogenic data. This may not be an issue in areas where land cover has remained stable for a long time, but it could affect applications in areas where human pressure has changed over the analysed time frame.

Authors: We concur with the Referee's concerns about a possible temporal mismatch between the temporal coverage of OLIGOTREND time series and some of the HydroATLAS attributes. By proceeding with the spatial join between OLIGOTREND stations and HydroBASINS, our objective was to demonstrate the great variability of eco-biogeophysical contexts in our database. To explore potential relationships between the temporal trends of water quality variables and catchment properties, we agree it would be better to rely on databases with the highest spatiotemporal match. However, there are currently no such products available for all the OLIGOTREND stations, and instead of using a mixture of non-harmonized catchment properties, we prefer to rely on the current reference at the global scale, i.e. the HydroATLAS. In the future, it will be up to

the users/developers of the OLIGOTREND database to make sure they cautiously join OLIGOTREND with other spatial databases in an attempt to understand the drivers behind the observed trends, an objective that goes beyond the current manuscript and for which the acknowledged temporal mismatch might not be that relevant.

In the revised version of the manuscript, we now insist that such a temporal mismatch between HydroATLAS variables and OLIGOTREND data may exist:

L142-146:

"Although we proceeded with the spatial join between HydroATLAS and OLIGOTREND stations, we acknowledge there may be a potential temporal mismatch between HydroATLAS properties and OLIGOTREND temporal coverage. Yet, we considered this spatial join would succeed at demonstrating the great variability of watershed and ecosystem properties encountered in the OLIGOTREND database."

R2C3. I also have concerns about the process used to link monitoring stations with HydroRivers. Even if only one river falls within 200 meters of a station, the resolution of the dataset does not guarantee that it is the correct river. You should consider adding flags to indicate the level of uncertainty in the river-station match. Ideally, this matching should be manually verified. However, as manual verification may not be feasible, comparing discharge values reported in HydroRivers with those reported by the station (when available) could help assess the uncertainty associated with each match.

Authors: We also share the Referee's concern here. The uncertainty inherent in this spatial join was already mentioned in the manuscript:

L154-155: "We acknowledge an important uncertainty for this step given the spatial resolution of the HydroSHEDS (15 arc-second)."

To go beyond this simple warning, we have identified for each OLIGOTREND river stations (n = 924) the 3 nearest river segments. For each possible match, we have calculated the distance between the station and each segment.

When the distance to the nearest segment was greater than 500 m, a flag was raised (flag = 1) indicating that the distance between the station and the segment might be too long to be considered a valid association. A dedicated warning was raised as follows: "Nearest segment is more than 500 m away".

When the distance to the 2$^{nd}$ (or 3$^{rd}$) nearest segment was comparable to the nearest segment (less than 10% of the distance to the nearest segment), a flag was raised (flag = 2) and a warning was produced: "There are several potential segments".

In that case,

- i) if we detected that other possible river segments pertained to several sub-basins (HYBAS_L12 in HydroATLAS documentation), this was indicated with a

flag set to 2.1, and a warning "Multiple potential segments. Other possible segments pertain to another catchment";

- ii) if we detected that other possible segments pertained to several drainage basins (MAIN_RIV in HydroRIVERS), the flag value was set to 2.2 and a warning was produced "Multiple potential segments. Other possible segments pertain to another river basin".

All other associations emerging from this spatial join were considered as valid, and flag value was set to flag = 0. We considered that only the stations with flag = 0 could be trusted.

Overall, out of 924 river stations, 90% were considered as valid (flag = 0). We calculated that 6.1% of stations were more than 500 m away from the closest HydroRIVERS segment (flag = 1), and 3.9% showed possible multiple associations (flag = or > 2), sometimes with different subbasins (1.3%, flag = 2.1) or drainage basins (0.3%, flag = 2.2).

[Figure]

This additional data analysis is now described in the revised version of the manuscript:

Lines 152-167:

"Finally, OLIGOTREND L1 river stations were linked to the RiverATLAS database by identifying the three nearest river segments using the function joinbynearest() in QGIS 3.26.2. For each possible station-segment match, the distance between the station and each segment was calculated, and the quality of the spatial join was assessed using a flagging system: if the distance to the nearest segment exceeded 500 m, a flag (flag = 1) was raised, indicating that the distance might be too large for the join to be considered valid. If the distance to the second or third nearest segment was less than 10% greater than the distance to the nearest segment, a flag (flag = 2) was raised indicating that several river segments could potentially be selected. In that case, if these segments were associated with multiple sub-basins (HYBAS_L12 in HydroATLAS documentation), a flag value of 2.1 was set. If these segments were linked to

multiple drainage basins (MAIN_RIV in HydroRIVERS), a flag value of 2.2 was set. All other associations identified during the spatial join were considered as valid, and flag value was set to flag = 0. Only stations with flag = 0 were considered reliable. Overall, out of 924 river stations, 90% were considered as valid. We found that 6.1% of stations were more than 500 m away from the closest HydroRIVERS segment, and 3.9% shown possible multiple associations (flag ≥ 2), sometimes with different sub-basins (1.3%, flag = 2.1) or drainage basins (0.3%, flag = 2.2). We acknowledge that there is some uncertainty in the spatial join between OLIGOTREND river stations and HydroRIVERS given the spatial resolution of the HydroSHEDS (15 arc-second). This uncertainty could be reduced by using a river network derived from a higher-resolution Digital Elevation Model."

This subsequent analysis reinforces our trust in the spatial join between OLIGOTREND river stations and HydroRIVERS. In the manuscript, Table 4 was updated by considering only the spatial associations with a flag = 0.

**Specific comments:**

R2C4: L98-100 Please specify what criteria was used (search terms) to find these datasets. How many papers/reports/books did you include?

Authors: We did not conduct a systematic review to gather all existing datasets. Instead, we prioritized harmonizing already known datasets as a first step towards a global database.

Yet, we interrogated the most common literature and data portals with the following search terms:

in TITLE or ABSTRACT (oligotrophication, reoligotrophication, chlorophyll, timeseries)

and in TITLE or ABSTRACT (lake, river, estuary, coastal, estuarine)

and in EVERYTHING (trend, long term, multi-decadal)

Out of this search in scientific literature databases, we found the following data sources:

| | |
|---|---|
| Naderian et al., 2024 | https://doi.org/10.1016/j.resconrec.2023.107401 |
| Lake PCI | https://doi.org/10.20383/102.0488 |
| Danish monitoring program | https://odaforalle.au.dk/login.aspx |
| Filazzola et al., 2020 | https://doi.org/10.1038/s41597-020-00648-2 |

From the EDI data portal, we found the following datasets:

LTER Florida Everglades
https://doi.org/10.6073/pasta/f45fbf88dcf1f78f0d74c1dbdaaa8c7d

Sacramento Bay Interagency monitoring
https://doi.org/10.6073/pasta/f58f8217c18f469e7fd565997a47813c

In the manuscript, we have edited Lines101-105 as follows:

"We then conducted a literature search on Web of Science (https://www.webofscience.com/wos/) and Scopus (https://www.scopus.com/) for further existing long-term *chla* and nutrient time series.To do so, we used the following search terms: TITLE or ABSTRACT (oligotrophication, reoligotrophication, chlorophyll, timeseries); and in TITLE or ABSTRACT (lake, river, estuary, coastal, estuarine) ; and in EVERYTHING (trend, long term, multi-decadal)."

R2C5: L100 Can you give a bit more context of why the database architecture allows "researchers to easily complement it with additional timeseries in the future". You can tell this by checking the shared repository, but a short text here would be good to understand a bit better strengths and the scope of the database.

Authors: We fully agree with the Referee that some information was missing. We have added the following sentence:

"New additions to the database will be facilitated by a set of scripts available in a dedicated version control GitLab repository (https://gitlab.com/OLIGOTREND/wp1-unify), allowing to reproduce, update or add more time series from level L0a to higher data levels and products." (Line 101)

R2C6: L138 What did you do when you found the "obvious mistakes"? The explained flags in L134 don't fit this category. Did you exclude these mistakes?

Authors: These obvious mistakes were directly corrected.

We have added this information to the text.

"Obvious mistakes in the units found in the original datasets at level L0b were identified and corrected by..."

R2C7: L151: Although you acknowledge the uncertainty associated with linking rivers and monitoring stations, I think you should flag this more explicitly. See my main comment above. Also, what did you do with stations that matched more than one river? Did you assign them manually?

Authors: Please see our response to comment R2C3 above.

R2C8: L231- 234 HydroATLAS land cover data is based on GLC2000, so it is unclear how you can determine associations for stations that do not match the timeframe of that land cover dataset.

Authors: Please see our response to comment R2C2 above.

---

## Author Response (AR3)

Dear Editor,

We are very pleased to see this work accepted for publication. We have revised the Supplementary file to ensure its title matches that of the main manuscript.
We have also updated the citation style and the list of references.
Finally, we have added a section for "Competing interests."

We are grateful for your time and look forward to seeing this data paper published soon.

Best regards,
Camille Minaudo, on behalf of all co-authors